# T-Cell Exhaustion in *Mycobacterium tuberculosis* and Nontuberculous Mycobacteria Infection: Pathophysiology and Therapeutic Perspectives

**DOI:** 10.3390/microorganisms9122460

**Published:** 2021-11-28

**Authors:** Andrea Lombardi, Simone Villa, Valeria Castelli, Alessandra Bandera, Andrea Gori

**Affiliations:** 1Infectious Diseases Unit, Foundation IRCCS Ca’ Granda Ospedale Maggiore Policlinico, 20122 Milano, Italy; valeria.castelli@unimi.it (V.C.); alessandra.bandera@unimi.it (A.B.); andrea.gori@unimi.it (A.G.); 2Department of Pathophysiology and Transplantation, University of Milano, 20122 Milano, Italy; simone.villa@unimi.it; 3Centre for Multidisciplinary Research in Health Science (MACH), University of Milano, 20122 Milano, Italy

**Keywords:** *Mycobacterium tuberculosis*, nontuberculous mycobacteria, immune checkpoint inhibitors, immune exhaustion

## Abstract

Immune exhaustion is a condition associated with chronic infections and cancers, characterized by the inability of antigen-specific T cells to eliminate the cognate antigen. Exhausted T cells display a peculiar phenotypic profile and exclusive functional characteristics. Immune exhaustion has been described in patients with *Mycobacterium tuberculosis* infection, and cases of tuberculosis reactivation have been reported in those treated with immune checkpoint inhibitors, drugs able to re-establish T-cells’ function. Exhausted T CD8^+^ cells’ profile has also been described in patients with infection due to nontuberculous mycobacteria. In this review, we initially provide an overview of the mechanisms leading to immune exhaustion in patients infected by *Mycobacterium tuberculosis* and nontuberculous mycobacteria. We then dissect the therapeutic perspectives related to immune checkpoint blockade in patients with these infections.

## 1. Introduction

Tuberculosis (TB) is an ancient and, therefore, extensively studied infectious disease caused by mycobacteria grouped in the *Mycobacterium tuberculosis complex*, among which *Mycobacterium tuberculosis* sensu stricto (hereafter *Mtb*) is the principal driver of the current global burden. The pathogen, acquired through inhalation, produces a disease localized in the lungs (i.e., pulmonary TB or PTB), although dissemination in virtually all organs (i.e., extrapulmonary TB or EPTB) is possible and accounts for a small fraction of cases [1].

Nontuberculous mycobacteria (NTM), also referred to as atypical mycobacteria, are a heterogeneous group of mycobacteria other than *Mtb* and *M. leprae*. They are environmental organisms that have long been considered nonpathogenic or responsible for disease only in immunocompromised individuals, affecting the pulmonary system in 90% of cases. Currently, their incidence is growing, ranging from 8.6 to 17.7 cases per 100,000 person-years, and has surpassed that of *Mtb* in Western countries [2,3]. Their ability to also cause disease in the immunocompetent has been acknowledged, and several risk factors have been identified, with pre-existing structural lung disease, genetically determined defects of cell-mediated immunity, oral corticosteroid treatment, chronic renal failure, and diabetes mellitus among the most relevant [4].

Both *Mtb* and NTM are intracellular pathogens that can produce a chronic infection requiring an efficacious and coordinated T-cell response to eliminate the microorganism. A mounting amount of evidence suggests that during chronic *Mtb* and NTM infection, as a consequence of chronic exposure to mycobacterial antigens and the cytokine milieu established, T cells of the host undergo a process of immune exhaustion (IE), which supports the continuation of the disease and abrogates the ability of the immune system to eliminate the microorganism.

In this review, we aim to provide an overview of the interaction between the immune system, *Mtb,* and NTMs, focusing on the condition of IE. After a brief description of what IE is and which mechanisms lead to it, we describe the current evidence on how IE relates to *Mtb* and NTM infection and possible therapeutic approaches aiming at reverting IE to increase pathogens’ clearance and, eventually, improve the clinical outcome (i.e., immune checkpoint inhibitors).

## 2. The Immunologic Background

The first evidence suggesting the existence of T cells specific for a certain pathogen, but unable to eliminate it, was provided by studies on chronic lymphocytic choriomeningitis virus (LCMV) infection in mice. Initially, Gallimore and colleagues showed how, in mice infected with LCMV, those who received high doses of the virus displayed a lower number of T cells directed against LCMV, which remained detectable for a lesser time, and with a reduced capacity of producing interferon (INF)-γ, compared to mice infected with low viral doses, suggesting that the exposure to a high amount of specific antigens can be detrimental to CTL function and survival [5]. During chronic LCMV infection, Zaiac and colleagues highlighted how CTLs recognizing a dominant epitope were deleted, whereas those recognizing minor epitopes expressed activation markers (CD69^hi^, CD44^hi^, CD62L^lo^) and proliferated in vivo, although they were unable to elaborate any antiviral effector function. This unresponsive phenotype was more pronounced under conditions of CD4^+^ T-cell deficiency, suggesting the importance of interaction between CD4^+^ and CD8^+^ lymphocytes to control chronic viral infections [6]. This dysfunction of virus-specific T cells occurs in a progressive manner, with an initial loss of IL-2 production, followed by tumour necrosis factor (TNF)-α, IFN-γ, and, eventually, the clonal deletion of antigen-specific CTLs [7]. Dysfunctional CTLs display inhibitory receptors on their surface, whose expression correlates with the severity of the CTL dysfunction [8]. Inhibitory receptors are key negative regulatory pathways that control autoreactivity and immunopathology [9]. Among them, we can enumerate programmed cell death protein-1 (PD-1), anticytotoxic T-lymphocyte-associated protein-4 (CTLA-4), T-cell immunoglobulin and mucin domain-containing protein-3 (TIM-3), T-cell immunoreceptor with Ig and ITIM domains (TIGIT), and lymphocyte-activation gene-3 (LAG-3). The blockade of these receptors by specific antibodies can partially restore the CTL function, as shown by Barber et al. in their study where dysfunctional CD8^+^ T cells were reinvigorated by blocking PD1 signalling during chronic LCMV infection [10]. Inhibitory receptors are not the only pathway involved in the dysfunction of T cells, which can be roughly classified into three general categories: (i) cell-to-cell signals, including prolonged T-cell receptor (TCR) engagement and costimulatory and/or coinhibitory signals; (ii) soluble factors such as excessive levels of inflammatory cytokines and suppressive cytokines, including interleukin-10 (IL-10) and transforming growth factor-β (TGFβ); and (iii) tissue and microenvironmental influences driven by changes in the expression levels of chemokine receptors, adhesion molecules, and nutrient receptors [11]. Pathogen-specific, dysfunctional CTLs have been identified in several infections occurring in humans, such as chronic hepatitis B [12], chronic hepatitis C [13], human immunodeficiency virus (HIV) infection [14], and sepsis [15].

Likewise, tumours, especially those caused by immunogenic cancers (e.g., melanoma, bladder cancer, and non-small-cell lung carcinoma), can generate dysfunctional CTLs as an immune escape mechanism [16]. In recent years, the introduction to therapy of molecules able to block immune checkpoints (i.e., immune checkpoint inhibitors (ICIs)), and thus partially revert CTL dysfunction, has revolutionized cancer immunotherapy, providing new hopes for previously hard-to-treat cancers [17].

Hence, T-cell exhaustion can be defined as a state of T-cell dysfunction that arises during many chronic inflammations, irrespective of whether the cause is infectious or tumoral. It is characterized by poor effector function, sustained expression of inhibitory receptors, and a transcriptional state distinct from that of functional effector or memory T cells, thus preventing optimal control of infections and tumours [18]. The existence of IE is likely linked to the large amount of chronic viral infections affecting humans, toward which we developed a sort of adaptation to prevent excessive immunopathologic tissue damage resulting from persistently infected tissues [19]. Some pathogens and tumours have allegedly evolved to take advantage of this mechanism to escape from host immune responses. The recently introduced ICIs have shown how it is possible to partially revert IE in cancer, paving the way for exploring the use of these drugs in other diseases.

## 3. Immune Exhaustion in MTB Infection

The primary form of host-pathogen interaction with *Mtb* does not automatically produce disease per se, but most likely, if the risk is not augmented by other endogenous and/or exogenous factors, it produces an infection without any clinical sign or symptom (i.e., TB infection), previously known as “latent” because it is thought to be sustained by metabolically “dormant” bacilli [20]. Tuberculosis infection can last for years, even decades, without producing active disease, thanks to a local immune balance able to contain mycobacteria inside lungs’ granulomas. Contrary to what one may think, TB infection is not a uniform and stationary state, but while acute TB infection progresses into its chronic phase, the quality of *Mtb* antigen recognition mutates (e.g., Ag85B antigen is more expressed during early phases of infection) [20]. Likewise, the *Mtb* mutation rate slows down, and the generation time becomes longer as long as the infection remains, thus explaining why TB reactivation is usually more frequent within the first 2 years after exposure. Subjects with TB infection during its “late latency” can, nevertheless, develop active disease, probably because of the exhaustion of the T-cell compartment, which, alone, however, cannot fully explain it [21].

Similar to what has been highlighted for chronic viral infections and cancer, chronic *Mtb* infection can lead to progressive impairment of *Mtb*-specific T-cell responses, which is inversely related to the mycobacterial load and can be partially restored during antimycobacterial therapy [22]. As expected by exhausted T cells, *Mtb*-specific CD4^+^ and CD8^+^ T cells ex vivo display reduced production of INF-γ, TNF-α, and IL-2. In particular, PD-1 and its ligands are widely expressed on T cells, monocytes, macrophages, and B cells of patients with active pulmonary tuberculosis [23,24,25]. The blockade of PD-1, or PD-1 and its ligands, can enhance the specific degranulation of CTLs and the percentage of specific IFN-γ-producing lymphocytes [22,26]. Interestingly, PD-1 knockout (KO) mice develop high numbers of *Mtb*-specific CD4^+^ T cells while displaying markedly increased susceptibility to infection; furthermore, PD-L1 KO mice also display enhanced, albeit less severe, susceptibility to infection [27]. In the same animal model, *Mtb*-specific T-cell proliferation is dramatically reduced in PD-1-deficient animals [28]. Overall, these data suggest that IE occurs during chronic active TB and could represent a mechanism employed by the immune system to avoid excessive immunopathology.

A key role in maintaining the infection is played by CD4^+^ T cells through the constant production of interleukin-12 (IL-12), which is responsible for keeping the differentiation of Th1 effector cells ongoing. The role of CD4^+^ T cells explains why HIV-infected, CD4^+^-depleted individuals are one of the vulnerable groups with an increased risk to develop TB. One of the roles of CD4^+^ T cells during TB infection is to orchestrate CD8^+^ T cells to kill *Mtb*-infected macrophages [29]. During the early phases of TB infection, CD4^+^ T cells promote the expression of several molecules in CD8^+^ T cells, such as INF-γ, TNF-α, IL-2, and chemokine receptors, to facilitate immune cell migration. Of note, INF-γ enhances the cooperation between dendritic cells and CD4^+^ T cells, whose interaction through the CD40-CD40 ligand is hindered by *Mtb* [30]. Besides triggering and enhancing CD8^+^ T-cell activity against *Mtb*, CD4^+^ T cells also prevent the exhaustion of CD8^+^ T cells during the chronic phase of TB infection. During this late stage, CD8^+^ T cells, if not supported by the CD4^+^ compartment, increasingly express coinhibitory molecules on their surfaces, such as PD-1, LAG-3, and TIM-3. The former binds PD-1 ligands (i.e., PD-L1 and PD-L2), which are expressed on *Mtb*-infected macrophages, neutrophils, and natural killer T (NKT) cells, leading to IFN-γ-induced cell lysis with, apparently, a negative feedback mechanism on NKT cell activation, thus preventing excessive tissue damage [27,31]. Therefore, the PD-1/PD-L pathway is not only involved in immunity mediated by *Mtb*-specific CD4^+^ T cells but also inhibits innate immunity (i.e., phagocytosis and antigen presentation) [23].

Another immune checkpoint involved in *Mtb* infection is TIM-3, which, similar to PD-1, is a negative regulator of T-cell function. Through interaction with galectin-9 (Gal-9)-positive macrophages, it leads to decreasing *Mtb* intramacrophage proliferation with IL-1β mediation [32]. The expression of TIM-3 can occur jointly with the expression of other makers, in particular with PD-1 (i.e., TIM-3^+^ PD-1^+^ CD4^+^ and CD8^+^ T cells), in the late stages of infection. This subset also coexpresses other receptors such as LAG3, which increase inhibitory cytokines (e.g., IL-10) and reduce pro-inflammatory ones (e.g., INF-γ, TNF-α, and IL-2) [33]. In contrast, the expression of CTLA-4, one of the most studied immune checkpoints, is limited to a small population of CD4^+^ T cells and is poorly expressed in CD8^+^ T cells during TB infection [33].

With global efforts oriented to ending TB, such as the strategy endorsed by the World Health Organization in 2015 [34], researching and developing effective TB vaccine(s) is a key instrument to succeed in it. In recent years, research has, therefore, focused on better understanding the pathophysiology and immune pathways at the root of *Mtb* intracellular survival and the progression of the infection. Several cells’ interactions and cytokines were discovered to participate in the pathophysiology of TB, including known immune checkpoints and, notably, the PD-1/PD-L1 pathway. Forthcoming strategies aiming to manipulate immune checkpoint pathways to modulate the immune response against *Mtb* are, therefore, highly plausible.

## 4. Immune Exhaustion in NTM Infection

Nontuberculous mycobacteria (NTM) are a cluster of almost 200 species and subspecies [35,36], many of which can be responsible for both pulmonary and extrapulmonary disease in humans, especially in those immunocompromised (with either primary or secondary immunodeficiencies) or with structural lung abnormalities, such as bronchiectasis, pneumoconiosis, chronic obstructive lung disease, and immunologic or genetic disorders (e.g., cystic fibrosis and primary ciliary dyskinesia). Several studies have suggested how the prevalence of pulmonary infections by NTM (i.e., NTM lung disease, NTM-LD) is increasing worldwide [37,38], as a consequence of improved diagnostic techniques and the growing number of at-risk individuals. The epidemiology of NTM-LD is not homogenous, with the *M. avium complex* (MAC) being the predominant isolate in North America and East Asia, whereas *M. kansasii*, *M. xenopi*, and *M. malmoense* are more common in Europe. Overall, NTM-LD is a chronic infection with persistent exposure of the host to antigens of the mycobacteria, intracellular pathogens, thus representing an archetypal condition for the development of IE.

Since NTMs have long been considered merely contaminants or colonizers, data are scarce regarding their interaction with the host’s immune system. Several key cytokines with a role in the immunopathogenesis of NTM-LD have nevertheless been identified. Kartalija et al. reported lower levels of IFN-γ and IL-10 after whole-blood stimulation with lipopolysaccharides (LPS) or live *M. intracellulare* in patients with NTM-LD compared to uninfected individuals with similar demographic characteristics [39]. Vankayalapati et al. [40], instead, reported higher concentrations of IL-10 and lower levels of INF-γ, IL-12, and TNF-α after stimulation of peripheral blood mononuclear cells (PBMCs) of patients affected by NTM-LD with heat-killed MAC. Overall, these findings suggest an imbalanced immune response toward NTM, specifically with inadequate production of IFN-γ, a key cytokine for the control of intracellular microbial infections.

Supporting the crucial role of IFN-γ in NTM-LD is the presence of a subgroup of individuals predisposed to develop the disease, resulting from mutations occurring in the IL-12/IFN-c/STAT1 signalling pathway, which led to impairment in IFN-γ signalling [4]. However, the subjects harbouring these mutations represent only a small minority of the total, and other mechanisms are likely responsible for reduced IFN-γ levels among those affected by NTM-LD. To better dissect this point, Shu et al. [41] in 2017 performed a prospective case-control study enrolling 50 patients with NTM-LD and 30 healthy donors (HDs). At baseline, cytokine production by PBMCs, PD-1 and PD-L1 expression, and apoptosis status on lymphocytes were investigated. PBMCs of all participants were collected and stimulated with heat-killed *M. avium* bacilli and MAC sensitin, showing higher TNF-α and INF-γ levels among HDs compared to those affected by NTM-LD. The expression of PD-1 and PD-L1 on PBMCs was higher in CD3^+^, CD4^+^, CD8^+^, CD19^+^, CD56^+^, and CD4^+^CD25^+^ cells in those affected by NTM-LD compared to HDs. Moreover, NTM-LD status was associated with higher early and late apoptosis status on CD4^+^ lymphocytes compared to HDs. As a second step, the authors investigated the effect of anti-MAC treatment in the same cohort. After two months of treatment, cytokine response was significantly increased, including TNF-α and IFN-γ, while PD-1 expression decreased slightly. This change in cytokine values suggests that chronic NTM infection can have a suppressive effect on PBMCs, and this is probably mediated through the interaction PD-1/PD-L1. To verify that, the authors treated PBMCs with anti-PD-1, PD-L1, and PD-L2 antibodies. After MAC stimulation, cytokine production significantly increased compared to PBMCs not pretreated with blocking antibodies. Moreover, flow cytometry analysis demonstrated decreased apoptosis levels in pretreated T lymphocytes, both CD3^+^ and CD4^+^, suggesting that blocking the PD-1 pathway might improve lymphocyte function and reduce apoptosis. Similar results were obtained with stimulation of the T lymphocytes compartment, after exposure to blocking antibodies, by MAC-primed macrophages. Overall, these results suggest that IE plays an important role in NTM disease pathophysiology, in which a decreased IFN-γ production is the final hallmark of this dysfunction.

The above-mentioned results were corroborated by findings recently provided by Han et al. [42], who investigated the characteristics of circulating CD4^+^ T-cell subsets in a subject affected by MAC pulmonary disease, performing a prospective case-control study on 71 NTM-LD individuals and 20 HDs. The authors stimulated PBMCs with heat-killed *M. avium* and *M. intracellulare* bacilli and evaluated CD4^+^ T-cell populations. They found a significant increase in the frequency of Th2 T cells and Tregs and a decrease in the frequency of Th17 T cells in NTM-LD individuals upon MAC stimulation compared to HDs. They also evaluated cytokines production related to the different T-cell subsets. In accordance with previous studies, IFN-γ, IL-10, and IL-17A production in response to MAC stimulation was significantly decreased in individuals with NTM-LD. Finally, they further highlighted the increased expression, in NTM-LD individuals after MAC stimulation, on CD4^+^ T cells of other immune checkpoints besides PD-1: CTLA-4 and TIM-3. This corroborates the idea that a suppressed T-cell-mediated response is associated with NTM-LD due to MAC.

Contrasting with the above-mentioned results are the data provided by Wu and colleagues, who detected no significant differences in the values of IFN-γ and TNF-α after stimulation with various mitogens and agonists, produced by various T-cell subsets, NK cells, or monocytes, between subjects with NTM-LD, disease control, or healthy controls [43]. Instead, they found lower levels of Th17 cells and IL-17A in NTM-LD individuals [43]. It should be underlined how only a minority (2/30, 6.6%) of subjects were treatment naïve, with 19/30 (63.3%) under therapy at the time of sampling, an important confounding factor that does not allow drawing solid conclusions about IE in this cohort. Nonetheless, the attenuated IL-17 response might contribute to host vulnerability or pathogen evasion via impairment of neutrophil recruitment and granulopoiesis [42]. Figure 1 provides a general overview of the IE process among patients with NTM-LD.

Additional data revealed that another pivotal lymphocyte population involved in NTM infection is Tregs. For example, CD4^+^CD25^+^Foxp3^+^ T cells are elevated upon TB infection, leading to the suppression of T-cell-mediated IFN-γ production. In fact, in IFN-γ KO mice infected with *M. massiliense*, progressive pulmonary disease and accumulation of Tregs in the lungs were observed [44]. Lutzky et al. [45] studied the immune profiles of two populations at risk for developing NTM infection, namely, the elderly and subjects with cystic fibrosis (CF). The latter, either with active or past NTM-LD, showed a T-cell phenotype with a deficiency in the TNF-α production after MAC stimulation, while in the former, a pattern of increased CD25 and CTLA-4 coexpression, as well as PD-1 expression, was observed on CD4^+^ T cells, suggesting an exhausted immune phenotype. The frequency of Tregs was significantly increased in both CF subjects, with both active and past NTM infection, and in the elderly compared to CF individuals with chronic *Pseudomonas aeruginosa* infection and healthy persons.

Overall, there is extensive evidence highlighting how IE is a condition that can be identified in patients with NTM-LD and how the blockade of PD-1 signalling can enhance IFN-γ production by CTLs, at least in the context of MAC infection. These data support the idea that IE could contribute to the pathophysiology of NTM-LD, thus suggesting the manipulation of IE as a possible novel therapeutic target in association with concomitant antimycobacterial therapy.

## 5. ICIs as Adjuvant in the Treatment of MTB/NTM Infection

Within the domain of TB, during recent decades, attention toward TB immune pathways and pathophysiology has expanded beyond the sole goal of vaccine research, leading to the development of host-directed therapies (HDTs), namely using molecules with the precise intention to alter the immune balance responsible for making some *Mtb* bacilli resistant to immune killing. The portfolio of HDTs for use in TB has expanded in recent years, although many pharmaceutical agents are still in the preclinical phases of the study. Only a few drugs (e.g., CC-11050, everolimus, auranofin, metformin, and ergocalciferol) have passed to the second phase of clinical trials [46]. Among HDTs investigated in TB, there are ICIs that act upon the previously described pathways. The use of ICIs as HDTs in TB infection and diseases warrants, however, caution due to possible excessive immunopathology [47]. Blocking PD-1 checkpoint, for example, by using nivolumab or pembrolizumab, two FDA- and EMA-approved humanized antibodies for cancer generally leads to TB progression into disease. Using these drugs was proven to reduce PD-1 expression in CD8^+^ T cells within granulomas in the lungs, while PD-L1 and CD8^+^ T cells remained highly expressed in the same setting [47]. This led to suggesting the use of the PD-1 blockade in adjunct to canonical TB preventive treatment (TPT) to “revitalize” metabolically dormant bacilli, increasing TPT sterilizing activity. The use of anti-PD-1 drugs in TB infection is, however, still mostly limited to nonhuman primate models [48], while conversely, literature on TB reactivation in cancer patients treated with PD-1 inhibitors is accruing [49]. The same reasoning and precautions should be applied for anti-PD-L1 drugs such as atezolizumab, avelumab, and durvalumab.

Unlike anti-PD-1, the TIM-3 blockade has been reported to restore T cells from exhaustion and reduce mycobacterial replication within macrophages, thus improving outcomes in TB infection. Using both anti-PD-1 and anti-TIM-3 ICIs was experimented with people living with HIV (PLHIV) [50] never exposed to antiretroviral therapy (ART), highlighting that the double blockade was able to restore macrophage and T-cell control over bacterial growth, including *Mtb* [50]. Namely, double-blocking PD-1 and TIM-3 receptors increased the secretion of IL-1β and promoted *Mtb* clearance [50,51]. Another double-blocking scheme might be of interest for the future, as low expression of LAG3 has been suggested to prevent TB progression [52], and the use of therapies inhibiting LAG3 and PD-1 has been shown to rescue CD4^+^ and CD8^+^ T-cell activity against another intracellular pathogen (i.e., *Plasmodium falciparum*) [53].

Research on ICIs to be used as new HDTs in both TB infection and disease must be, regardless, oriented to specific objectives as using either PD-1 or PD-L inhibitors, and in general, ICIs, as adjunctive therapy to TPT, should be limited to those highly at risk of TB progression, as well as adjunctive treatment, aiming for the eradication of *Mtb* host reservoirs, in drug-resistant TB, such as multidrug-resistant (MDR) and extensively resistant (XDR) forms [54], and in fragile populations (e.g., PLHIV).

As mentioned above, the PD-1 pathway is overexpressed and involved in the pathophysiology of NTM infection. Similar to what has been postulated for *Mtb* infection, the use of PD-1 (or CTLA-4) antibody blockade therapy has also been hypothesized in this setting [55]. Currently, however, no in vivo assessment has yet been performed. Instead, a growing amount of evidence showing the possibility of NTM infection reactivation during treatment with ICIs is accruing. Recently, Ananad et al. described thirteen cases of NTM infection during ICI therapy: nine due to nivolumab, two due to pembrolizumab, one due to durvalumab, and one due to atezolizumab. Several other cases of NTM infection during ICIs treatment for lung cancer have been reported in the literature [56,57]. It should be underlined that in all of the described cases, patients were previously treated with standard chemotherapy.

In 2018, a controversial case of advanced non-small-cell lung carcinoma in which NTM disease improved after nivolumab administration was described [58]. In this report, a 73-year-old Japanese man was diagnosed with advanced lung adenocarcinoma and underwent six cycles of standard chemotherapy. After one year, concomitantly with the worsening of the clinical condition, he was diagnosed with NTM-LD by *M. abscessus* subsp. *massiliense*. After two weeks of antibiotic treatment with imipenem and amikacin, clinicians halted the administration of antibiotics and started the administration of nivolumab because of tumour relapse. NTM-LD improved during nivolumab treatment. Only one negative cultural exam on the sputum was performed to confirm NTM-LD recovery. It should be noted that, in some cases, successful treatment can be achieved with source control alone [59]. Table 1 summarizes the cases reported in the literature of NTM-LD that occurred in patients undergoing immunotherapy for cancer.

## 6. Conclusions

Overall, IE appears as a mechanism preserved during evolution, which has probably been conserved to avoid excessive immune response against microorganisms that are difficult to eradicate. In both *Mtb* and NTM infection, IE features can be observed in the T-cell compartment and can be partially abrogated through the administration of molecules blocking the immune checkpoint receptors. The observations that KO PD-1/PD-L1 mice can have a markedly increased susceptibility to infection, and the anecdotal observations of *Mtb* and NTM infection reactivations in patients receiving ICIs, suggest caution about their employment in these groups of patients. Nonetheless, targeted regulation of the PD-1 pathway may have therapeutic potential for both TB and NTM-LD in the future, probably in combination with canonical antimicrobial therapy. This might be of interest to vulnerable subjects or hard-to-treat mycobacteria. Fine-tuning, instead of complete abrogation, should be the ideal approach for PD-1 modulation during these diseases’ treatment.

## Figures and Tables

**Figure 1 microorganisms-09-02460-f001:**
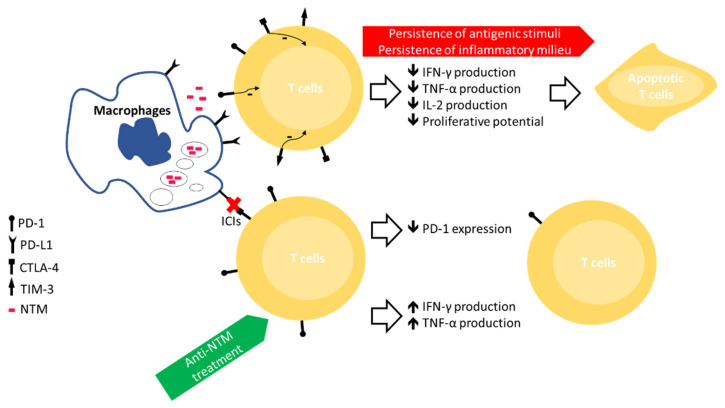
Overview of the immune exhaustion process involving T lymphocytes in subjects with NTM-LD and the possible approaches to overcome it. ICIs, immune checkpoint inhibitors; PD-1, programmed cell death protein-1; CTLA-4, anti-cytotoxic T-lymphocyte-associated protein 4; TIM-3, T-cell immunoglobulin and mucin domain-containing protein 3; PD-L1, PD-1 ligand 1.

**Table 1 microorganisms-09-02460-t001:** Overview of studies assessing the impact of ICIs in MTB/NTM infection.

	Age/Gender	Tumour	ICI	Prior Radiotherapy or Chemotherapy	NTM Species Identified	Treatment for NTM Infection
Fujita et al. 2020 [56]	78 years, Female	Lung adenocarcinoma	Nivolumab	Standard chemotherapy	*M. intracellulare*	MAC treatment + nivolumab
Fujita et al. 2020 [56]	80 years, Male	Non-small cell lung cancer	Atezolizumab	Radiotherapy + standard chemotherapy	*M. avium + M. intracellulare*	MAC treatment + atezolizumab
Fujita et al. 2020 [56]	66 years, Male	Lung squamous cell carcinoma	Nivolumab + Atezolizumab	Standard chemotherapy	*M. intracellulare*	No medication for severe debilitation
Baba et al. 2020 [60]	80 years, Male	Lung squamous cell carcinoma	Durvalumab	Radiotherapy + standard chemotherapy	*M. avium*	-

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
