# Peer review of "T-Cell Exhaustion in Mycobacterium tuberculosis and Nontuberculous Mycobacteria Infection: Pathophysiology and Therapeutic Perspectives"

_microorganisms, 2021, doi:10.3390/microorganisms9122460_

Round 1

Reviewer 1 Report

In the article under review the authors analyze a very interesting and important for the disease outcome phenomenon immune exhaustion during chronic mycobacterial infections. The manuscript is well structured. The manuscript details the publications of recent years in this scientific area. The authors' original view of this problem is visible. Both the causes of exhaustion and the ways of solving this problem are analyzed in detail. This manuscript cold be recommended for publication as it is.

Reviewer 2 Report

Excellent review article. In my opinion it is well structured and objective. I would like to point to the authors that in cases of NTM some patients clear infection on their own after adequate source control and this should be acknowledged ( https://www.sciencedirect.com/science/article/pii/S2214250921002882)
